# Reduced activity, reactivity and functionality of the sympathetic nervous system in fibromyalgia: An electrodermal study

**Gustavo A. Reyes del Paso, Pablo de la Coba** * 

Department of Psychology, University of Jaén, Jaén, Spain

* pcoba@ujaen.es

**Data Availability Statement:** The research data of the study are available to the public via the repository Open Science Framework (OSF: https://osf.io/3uk4e/).

## Abstract

Alterations in autonomic activity are well established in fibromyalgia syndrome (FMS). Previous studies found reduced parasympathetic activity and sympathetic reactivity to physical and stress manipulations. However, sympathetic activity at rest has not been well studied in FMS. Sweating is exclusively controlled by sympathetic mechanisms. In this study, skin conductance (SC), as an indirect measure of sweating, was analyzed in 45 women with FMS and 38 healthy women. Tonic SC levels were recorded during a 4-minute rest period, and a breathing maneuver consisting of deep breathing with posterior breath holding was used to evoke SC responses. Associations of tonic SC with state anxiety and body temperature, measured in the hand, were explored to determine sweat functionality. The results showed reduced tonic SC levels, with a less marked decrease in SC during the recording period, and blunted SC reactivity to the breathing manipulation in FMS patients relative to healthy participants. Positive associations of SC with state anxiety and body temperature were observed in healthy participants, but these associations were absent in FMS patients. These results indicate alterations of sweating in FMS, suggesting reduced tonic and reactivity sympathetic influences. Furthermore, the absence of associations between SC levels and state anxiety and body temperature in the patient sample suggested a loss of functionality of the autonomic nervous system in FMS. Diminished autonomic regulation in FMS would reduce the ability to cope with environmental demands, thus favoring increases in stress and pain levels. Finally, the observed reduction in sweating is in accordance with evidence of small nerve fiber neuropathy in FMS.

## Introduction

Fibromyalgia syndrome (FMS) is one of the most prevalent chronic pain disorders. It is accompanied by symptoms such as fatigue, insomnia, negative affect (anxiety and depression), cognitive deficits, etc. [1]. The etiology and pathophysiology of FMS are largely unknown, and there are no specific somatic signs of the disease. Most etiological models propose that FMS pain arises from central nervous system sensitization [2–4] and/or peripheral small fiber neuropathy [5, 6].

Clinical and experimental studies have revealed autonomic nervous system (ANS) deficiencies in FMS [7–12]. One of the most systematically studied functions of the ANS in FMS is

**Funding:** This research was supported by a grant from the Spanish Ministry of Science, Innovation and Universities, co-financed by FEDER funds (Project RTI2018-095830-B-I00).

**Competing interests:** The authors have declared that no competing interests exist.

cardiovascular regulation. At rest, FMS patients show increased heart rate, decreased heart rate variability (HRV) and baroreflex sensitivity (BRS), and reduced stroke volume and myocardial contractility [15–18]. In response to stress manipulations, blunted cardiovascular reactivity, expressed in terms of heart rate, blood pressure, BRS, stroke volume and myocardial contractility, has been observed [12, 17–19]. HRV and BRS are markers of cardiac parasympathetic activity [20], while stroke volume and myocardial contractility, mainly controlled by β1-adrenegic influences, are sympathetic markers [21]. This evidence suggests reduced parasympathetic and sympathetic autonomic activity and reactivity in FMS.

Some studies have also reported autonomic abnormalities in FMS, both at rest and under stress conditions, in terms of skin conductance (SC), temperature and blood flow, although with conflicting results. For example, FMS patients showed higher SC reactivity to emotional and mental stress than healthy individuals, with no differences seen at baseline [8, 9, 22, 23]. This greater SC reactivity contrasts with findings observed in other chronic pain conditions (e.g., chronic low back pain), in which lower SC reactivity was observed [24]. Regarding peripheral blood flow, decreased microcirculatory blood flow (indicating greater vasoconstriction) has been observed in tender points of FMS patients [25]. Blood flow dysregulation has been implicated in the dysfunctional thermoregulation observed in FMS patients [11]. Given that SC and vascular tone are sympathetically mediated, these results would suggest enhanced sympathetic responses to stress in FMS. However, discrepant results have also been reported, including observations of compromised sudomotor function [26], and weaker skin temperature and blood flow responses to intense auditory stimulation and the cold pressor test [27], suggesting reduced sympathetically mediated vasoconstriction.

It has been suggested that dysregulation of the ANS and, specifically an overactive sympathetic nervous system (SNS), might be important to understand chronic pain and FMS [28, 29]. Specifically, it has been suggested that abnormalities in the SNS may be involved in the pathogenesis of some pain disorders, especially neuropathic pain. Based on the hypothesis of an overactive SNS, it has been proposed that FMS is a sympathetically maintained pain disorder [19]. Additionally, some authors have suggested that the fatigue and widespread pain characterizing FMS might be secondary to peripheral tissue ischemia produced by excessive vascular tone, due to sympathetically mediated vasoconstriction [30]. Extreme sensitivity to cold and stress conditions, and the associated vasospasms [31], may be in accordance with this hypothesis.

The activity of the eccrine sweat gland is exclusively controlled by the SNS. Postganglionic non-myelinated C-fibers from the sympathetic ganglia combine with peripheral nerves and travel to sweat glands, which interlace the periglandular tissue with cholinergic terminals [32]. Thus, any increase in SNS activity can cause sweating. Sweat has two main functions: thermoregulation via evaporative heat loss, and adaptation to environmental demands by modulating humidity in the hands and feet, thus improving the manipulation and grasping of objects, and helping to prevent injures [32, 33].

One method to evaluate small fiber neuropathy, a hypothesized pathophysiology in FMS, is the measurement of sweating via quantitative sudomotor axon reflex testing [34] or distal electrochemical SC [26]. Thus, sweating might also provide insight into SNS innervation of the skin.

Sweating is measured in psychophysiology via electrodermal activity, traditionally taken as an indicator of emotional and attentional processes [33]. More extensive sweating facilitates the conduction of electricity through the skin. Two main variables are obtained from SC, levels (tonic activity in the absence of stimulation) and responses. In this study, SC was measured in FMS patients and a sample of healthy participants. To measure the SC levels, we used a rest period, in which a progressive decrease in SC is typically observed, indicating adaptation to a

new situation (in this case the laboratory). Previous studies in FMS patients have evaluated SC responses to emotion, stress or arousing procedures [8, 9, 22, 23]. FMS patients are more prone to suffering from stress [8, 9, 13], which would facilitate greater SC responses to these specific stimuli. In this study, we analyzed physiological-related SC reactivity. In order to elicit SC responses, a breathing maneuver consisting of deep breathing with posterior breath holding was used. Deep breathing is a powerful stimulus for general autonomic activation, and for stimulation of autonomic reflexes [35, 36]. To analyze the functionality of sweating, we evaluated associations of SC with state anxiety (i.e., adaptation to environmental requirements; greater anxiety levels produce increases in SC) and body temperature (i.e., thermoregulation; higher body temperature increases SC). Finally, at an exploratory level, associations between SC variables and levels of pain and fatigue were analyzed.

Based on previous reports of reduced and dysfunctional ANS activity at rest, and blunted autonomic responses to physical manipulations [12,17–19], we predicted the following features in FMS patients (in comparison with healthy participants): (1) reduced SC levels at rest, (2) reduced decrease in SC (i.e., physiological adaptation) during the rest period, (3) blunted SC responses to deep breathing, and (4) less marked modulation of state anxiety and body temperature on SC levels during the rest period, leading to weaker positive associations of SC with state anxiety and body temperature. Finally, we predicted inverse associations between SC variables and levels of pain and fatigue.

## Method

### Participants

Given the greater female prevalence observed in FMS [1], and in order to avoid gender-related differences, the study enrolled only females. The participants were 45 women diagnosed with FM by a rheumatologist according to 1990 American College of Rheumatology criteria [1] and recruited from the Fibromyalgia Association of Jaén. The 1990 diagnostic criteria were used to unify diagnosis given that several patients have the FMS diagnosis before 2010. Thirty-eight healthy women recruited from the community via local advertisements and snowball sampling were included in the control group. Inclusion criteria for all participants were being a woman between 20 and 72 years old with a basic educational level to ensure comprehension of instructions and self-reported instruments. Exclusion criteria for all participants were any kind of cardiovascular and autonomic disease, metabolic abnormality, skin or neurological disorder (including brain injury or cerebrovascular events), drug abuse, or severe somatic (e.g., previous malignant diseases) or psychiatric (e.g., psychotic, bipolar or obsessive-compulsive) disorders. Additional exclusionary criteria for the control group were the suffering of a pain disorder of any kind, and to have a first-degree relative diagnosed with FMS. However, individuals with transient acute pain (e.g., menstrual pain, occasional head, muscle or gastrointestinal pain) were allowed to participate. Recruitment was performed by means of a brief telephone interview to each possible participant in which basic information was required (age, basic education level and medical information such as other pathologies or medication use) and short instructions for the participation were given. The study was approved by the Ethics Committee of the University of Jaén with ethical code reference "OCT.18 / 1.PRY" and date of approval November 2nd 2018. All participants signed a written informed consent that was previously presented in detail. The data for the variables of interest are displayed in Table 1.

Most of patients take different combinations of medications. Eleven patients take antidepressants, anxiolytics and non-opioids analgesics; 6 take anxiolytics and non-opioids analgesics; 5 take antidepressants, anxiolytics, non-opioids analgesics and opiates; 5 take both opiates and non-opioids analgesic; 4 patients do not take any king of medications; 4 take anxiolytics

**Table 1. Demographic, clinical, medication use (number of participants and percentage in brackets) and physiological data in fibromyalgia (FMS) patients and healthy participants.**

| | FMS (n = 45) | Healthy (n = 38) | t or $\chi^2$ | p | η2 |
|---|---|---|---|---|---|
| Age (years) | 51.53±8.89 | 50.08+10.22 | .693 | .490 | .006 |
| BMI | 27.26±4.76 | 26.17+3.64 | 1.184 | .240 | .016 |
| State Anxiety (STAI) | 21.87±12.16 | 13.00±7.47 | 4.07 | < .001 | .159 |
| N˚ of pain points (MPQ) | 21.98±6.76 | 4.00±3.84 | 15.17 | < .001 | .722 |
| Sensorial pain (MPQ) | 32.11±21.73 | 13.00±7.33 | 5.54 | < .001 | .248 |
| Total MPQ Score | 49.29±30.67 | 20.45±11.35 | 5.85 | < .001 | .271 |
| Current pain intensity (MPQ) | 2.62±1.27 | 0.84±0.79 | 7.81 | < .001 | .411 |
| Fatigue (FSS) | 49.42±11.31 | 20.66±13.48 | 10.57 | < .001 | .580 |
| Antidepressant use (%) | 23 (51.11) | 3 (6.67) | 17.89 | < .001 | .215 |
| Anxiolytic use (%) | 29 (64.44) | 7 (15.56) | 17.768 | < .001 | .214 |
| Analgesic use (%) | 36 (90) | 4 (8.89) | 39.828 | < .001 | .480 |
| Opiate use (%) | 19 (42.22) | 0 (0) | 20.808 | < .001 | .251 |
| SC level (μS) | 2.02±2.11 | 3.56 ±2,11 | -2.15 | .034 | .034 |
| Slope SC (μS/m) | -0.153±.237 | -0.344±.584 | 2.01 | .048 | .048 |
| Hand Temperature (Cº) | 35.91±1.03 | 35.92± .82 | -.039 | .969 | .969 |

**Note:** Results of group comparisons are also reported (Student *t* test or $\chi^2$). Effect size was reported as adjusted eta squared (**η2**). MPQ = McGill Pain Questionnaire; FSS = Fatigue Severity Scale.

and opiates and non-opioids analgesic; 3 take antidepressants, anxiolytics, and opiates; 3 only non-opioids analgesics; 2 antidepressants and opiates; and 2 antidepressants and non-opioids analgesics.

## Physiological recording

A Biopac MP36 polygraph (Biopac Systems Inc., Goleta, CA) with Acknowledge 4.2 software was used for data acquisition and signal preprocessing. SC was recorded in μSiemens (μS) via the Biopac SS57LA lead set with a constant current of 0.5 V. Recording was performed through two disposable Ag-AgCl isotonic gel electrodes (Biopac EL507) with a 1 cm$^2$ contact area placed on the thenar and hypothenar areas of the right hand. Hand temperature in degrees Celsius (C˚) was recorded using a Biopac TSD202A transducer consisting of a small thermistor (1.7mm x 5mm) placed in the thenar area of the right hand. A hypoallergenic sticking plaster was used to fix the thermistor to the hand. Physiological recording was performed at 200 Hz.

## Procedure

The study was performed during one approximately 50-minute session. A clinical psychologist recorded the clinical history, checked the inclusion-exclusion criteria and obtained written informed consent from all patients. Then, the McGill Pain Questionnaire (MPQ) [37] was presented. Four parameters were obtained from this instrument: a) the number of pain points marked on the picture of the human body; b) the sensorial component of pain (scale range: 0–84); c) the global MPQ score (scale range: 0–146); and the "current pain intensity index" (scale range, 0–5). Values of Cronbach´s α between 0.56 and 0.74 were reported for this instrument [37]. Furthermore, the Fatigue Severity Scale (FSS) [38] was also presented in order to dispose of a measure of habitual fatigue (score range: 9–63; Cronbach´s α: .88). Finally, the state subscale of the Spanish version of the State-Trait Anxiety Inventory [39] was administered. This subscale assesses current (at the time of test administration) anxiety levels using a

4-point Likert scale (score range: 0–60; Cronbach´s α: .93) [39]. Physiological recording took place in a dimly lit, sound-attenuated room with a temperature maintained at 22˚C. The palm of the right hand was cleaned, and the electrodes were fixed. During the 6.5-min rest period, participants were instructed to relax in a sitting position with their eyes open, and to refrain from speaking. The first 2.5 minutes was a stabilization period, during which data were not recorded; the following 4 minutes of the rest period were analyzed. Recording started approximately 11 minutes after the participant arrived at the laboratory. After this period, the experimenter trained the participant to take a quick and deep breath, and to hold it for 6s. Thus, the breathing manipulation required rapid, deep and sustained inspiration. The training lasted until the deep breaths and holding of the air in lungs were carried out in a semi-automatic way, and in a similar manner among all participants. Afterwards, the participants were asked to take two deep breaths with an inter-breath interval of 1-min. All participants were asked not to consume any analgesic drugs, alcohol, or caffeine, and not to engage in intense physical exercise for 24 hours before the session. In each group half of participants performed the study in the morning and half in the afternoon, in both cases after 1.5–2 hours of having breakfast or lunch.

In some participants, the thermistor detached from the skin leading to unreliable temperature data. For this reason, the hand temperature data of only 36 FMS patients and 33 healthy participants were available for the analysis. Furthermore, some participants did not learn to perform the breathing maneuver in a standardized manner; the data sample for the SC response to deep breathing comprised the 39 FM patients and 28 healthy participants who carried out the breathing task correctly.

## Data reduction and Statistical analysis

In addition to the overall SC average, SC values were taken at seconds 0, 30, 60, 90, 120, 150, 180, 210 and 240 during the 4-min rest period. Hand temperature was averaged over the entire period. The amplitude of the SC responses to the breathing manipulation was obtained as the difference between the peak maximum SC response and the average value over the 5 s prior to the SC increase. The average amplitude of the two breathing trials was taken as the dependent variable.

The time course of SC was analyzed by ANOVA with one between-subjects factor (Group: FMS patients vs. healthy participants) and one repeated-measure factor (the nine SC values). In order to evaluate the effect of state anxiety and hand temperature on SC changes in each group, these variables were included separately as covariates in further ANCOVA models. The Greenhouse-Geisser procedure was used to correct for degrees of freedom. Results are reported with the original degrees of freedom and the corrected p values. Effect sizes are reported as eta-squared ($\eta_p^2$). Group comparisons were performed using Student's t tests for independent samples. Associations between variables were analyzed using Pearson correlation analysis.

## Results

### Levels of skin conductance

Fig 1 displays the changes in SC during the 4-minute recording period. SC progressively decreased (effect of Time: $F(8, 648) = 19.85$, $p < .0001$, $\eta_p^2 = .197$), but as a function of Group (interaction Group x Time: $F(8, 648) = 4.11$, $p = .038$, $\eta_p^2 = .048$). Although the effect of Time was significant in both groups (ps $< .01$, $\eta_{ps}^2 > .2$), the decrease in SC was greater in healthy participants than FMS patients. The slope of the SC decrease (see Table 1) was significantly greater in healthy participants than FMS patients. Finally, SC levels were lower overall in FMS patients

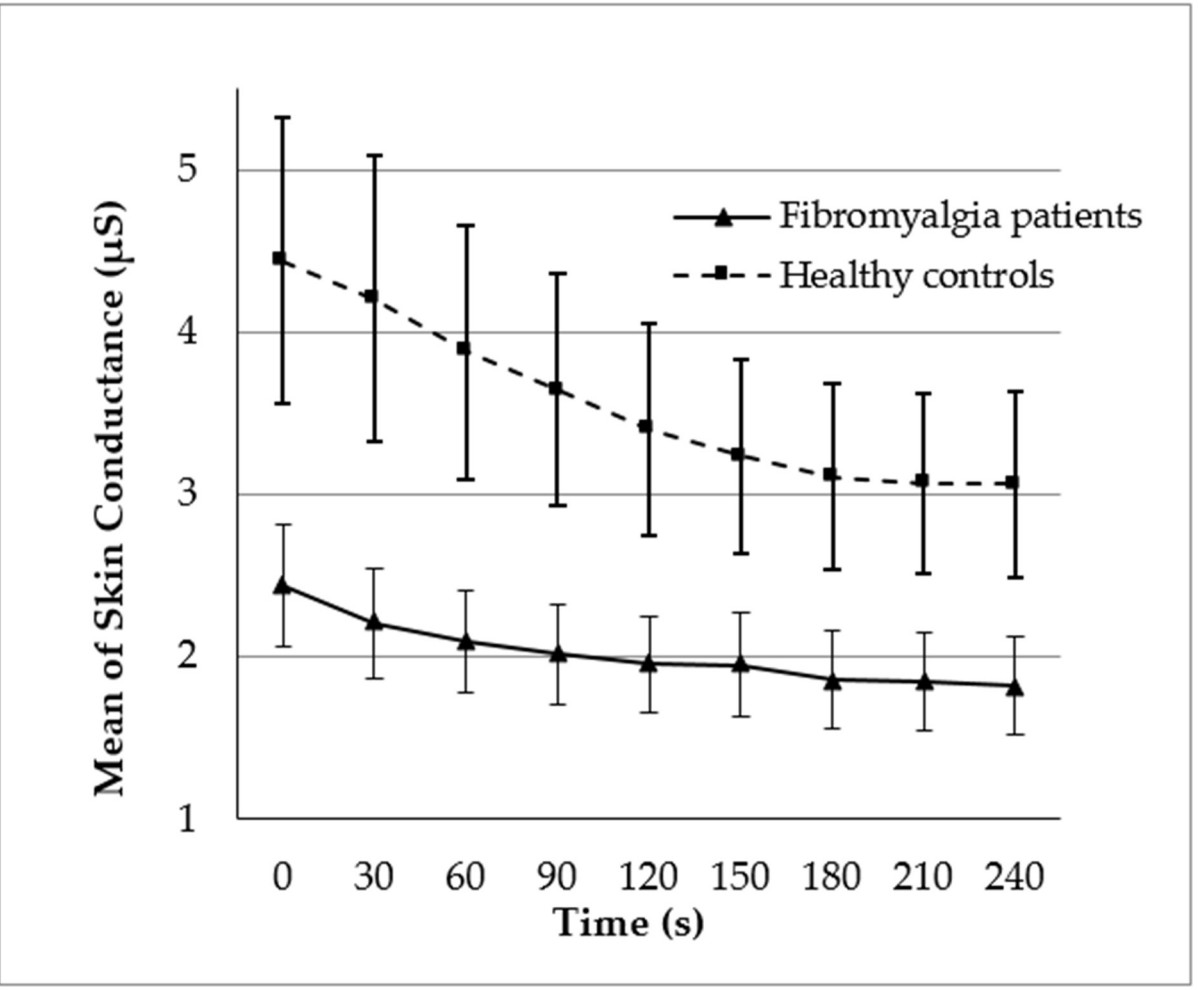

**Fig 1. Skin conductance levels during the 4-min recording period; values were obtained at 30-second intervals.** Bars indicate standard errors of the mean.

than in healthy participants (effect of Group: $F(1, 81) = 4.63$, $p = .034$, $\eta_p^2 = .054$). When state anxiety was included as a covariate, separately for each group, its main effect $F(1, 36) = 6.83$, $p = .013$, $\eta_p^2 = .159$) and interaction with Time ($F(8, 288) = 8.26$, $p = .005$, $\eta_p^2 = .187$) were significant in healthy participants. In this group, greater state anxiety was associated with higher SC levels and greater SC decreases during the recording period. When mean hand temperature was included as a covariate, its main effect was significant $F(1, 31) = 4.65$, $p = .039$, $\eta_p^2 = .130$) in healthy participants, and its interaction with Time was marginally significant ($F(8, 248) = 3.41$, $p = .068$, $\eta_p^2 = .099$). In healthy participants, higher hand temperature was associated with higher SC levels, and marginally associated with greater SC decreases (see below for a further description of these results). Neither of these effects were significant in FMS patients (all ps > .7).

### Skin conductance response to deep breathing

Fig 2 displays the SC response to the breathing maneuver as a function of group. The amplitude of the SC response was lower in FMS patients than in healthy participants ($t(65) = -2.44$, $p = .018$, $\eta_p^2 = .084$).

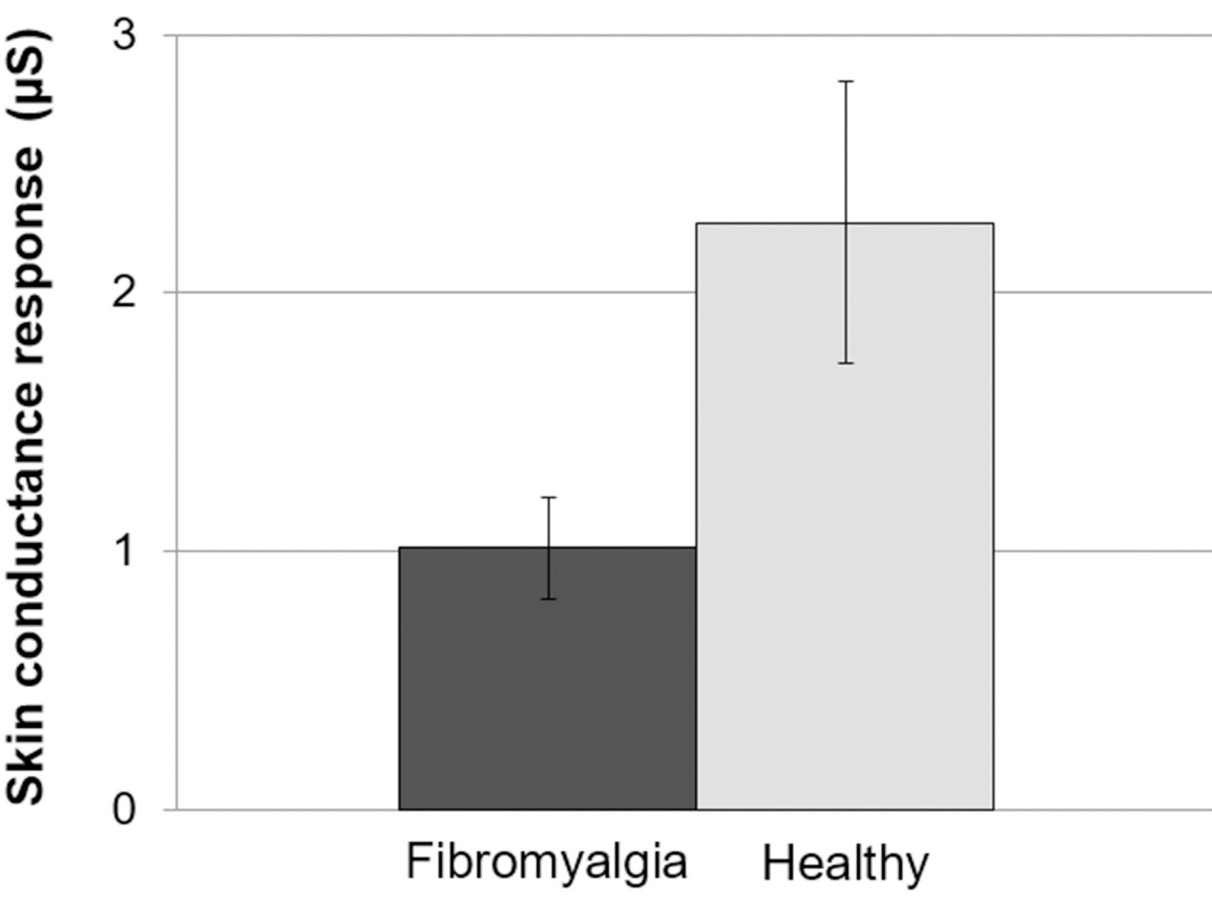

**Fig 2. Amplitude of the skin conductance response to the deep breathing maneuver.** Bars indicate standard errors of the mean.

## Associations among skin conductance, state anxiety and temperature

In healthy participants, overall mean SC levels were positively associated with state anxiety and mean hand temperature (see Table 2). State anxiety was also positively associated with hand temperature and negatively associated with the slope of the decrease in SC. Finally, hand temperature was marginally associated with the slope of the decrease in SC ($p = .059$). In FMS patients all of these correlations were non-significant (see Table 2).

## Associations among skin conductance and levels of pain and fatigue

Table 3 displays correlations between SC variables and measures from the MPQ and FSS in the whole study sample. Both mean SC during the rest period and the SC response to the breathing

**Table 2. Correlations between mean skin conductance (SC), slope of the SC decrease during the rest period, state anxiety, and mean hand temperature (T C˚) in the two study groups.**

| Variables | Healthy individuals | | Fibromyalgia patients | |
|---|---|---|---|---|
| | **State Anxiety** | **T C˚** | **State Anxiety** | **T C˚** |
| Mean SC | 0.40* | 0.36* | 0.05 | 0.18 |
| Slope SC | -0.44** | -0.33 | 0.01 | -0.03 |
| Mean T C˚ | 0.44* | – | -0.04 | – |

*$p < .05$
**$p < .01$

**Table 3. Correlations between mean skin conductance (SC) and SC response and variables from the McGill Pain Questionnaire (number of pain points, sensorial pain, total McGill score and current pain intensity) and Fatigue Severity Scale (fatigue) in the whole sample.**

|  | Mean SC (n = 83) | SC Response (n = 67) |
|---|---|---|
| N˚ of pain points | -0.17 | -0.27* |
| Sensorial pain | -0.22* | -0.23 |
| Total McGill score | -0.19 | -0.22 |
| Current pain intensity | -0.33** | -0.32** |
| Fatigue | -0.24* | -0.29* |

* p < .05
** p < .001

maneuver were inversely associated with fatigue (FSS) and current pain intensity (MPQ). Mean SC was inversely associated with sensorial pain (MPQ) and the SC response was inversely associated with the number of pain points (MPQ). Associations with the total MPQ score did not reach significance (p = 0.085 for mean SC and p = 0.079 for SC response).

## Discussion

Our results showed both reduced tonic SC levels and SC responses in FMS patients in comparison to healthy participants. Furthermore, positive significant associations of SC levels with state anxiety and body temperature were observed in healthy participants, but these associations were absent in FMS patients.

The reduced SC levels in the FMS patient sample indicated lower levels of sweating. Given that sweating is controlled exclusively by sympathetic cholinergic terminals, this suggests reduced tonic SNS influences. This result is in accordance with reports of reduced sudomotor activity in FMS [26], although other studies found no differences in resting SC levels [8, 9]. Not only were the average SC levels lower in the patient sample, but the slope of the decrease in SC throughout the rest period was also reduced in comparison to healthy participants. This suggests slower adaptation to the evaluation situation (laboratory sitting), in turn suggesting fewer autonomic adaptation resources and less flexibility in response to environmental changes. This lower ability to adapt to situational requirements has previously been observed in FMS and general chronic pain disorders. For example, chronic pain patients showed an aberrant pattern of autonomic cardiovascular responses to orthostatic changes (both to standing and lying), with this altered response being indicative of disease severity [12, 40]. In the cognitive domain, FMS patients showed less capacity to adapt and adjust to task requirements, resulting in reduced cognitive performance [41].

Regarding SC responses, most previous studies indicated greater reactivity to stress or arousing manipulations [8, 9, 22, 23]. However, in other chronic pain conditions [24] and functional disorders [42], reduced SC reactivity was observed. FMS patients report high rates of stressful live events and increased feelings of stress [8, 9, 13, 14], which in turn would mediate greater SC responses to emotional stimuli [8, 9, 22, 23]. In order to avoid this influence, we measured SC reactivity to a physiological manipulation known to evoke a potent autonomic response. Deep breathing with posterior air holding is a powerful stimulus for general autonomic activation, including a significant SC response [35, 36]. The response to this breathing maneuver was lower in FMS patients than in healthy participants, suggesting blunted physiological-related autonomic reactivity.

These results are in accordance with previous observations regarding autonomic cardiovascular control in FMS. Reduced myocardial contractility (mediated by $\beta_1$-adrenegic activity) at rest [17, 18], and blunted reactivity in cardiac sympathetic-mediated parameters to orthostatic stress and the cold pressor test, have been reported [12, 18, 19]. Taken together, these results suggest both reduced tonic sympathetic influences and blunted sympathetic reactivity in FMS.

To determine the functionality of sweating, we analyzed modulation over SC levels exerted by two factors known to increase sweating: state anxiety and body temperature. Furthermore, associations of SC levels with state anxiety and body temperature were computed. As expected, in healthy participants ANCOVA showed that state anxiety modulated SC levels and SC decreases during the recording period. Body temperature also modulated SC levels and marginally SC decreases during the recording period in healthy participants. None of these modulatory effects arose in FMS patients, however. Confirming these results, correlation analysis showed that levels of SC were positively associated with state anxiety and body temperature in healthy participants, but correlations were far from being significant in the patient sample. This absence of significant correlations was surprising and suggests a loss of function of sweating in FMS. On the one hand, increases in state anxiety stimulated SNS activity. In order to adapt to environmental stimulations and challenges (which increase state anxiety), the SNS increases sweating, among many other effects [33]. On the other hand, when body temperature increases, sweating is elicited in order to reduce temperature through the evaporation of sweat. According to our results, both functions appear to be aberrant in FMS patients. Furthermore, state anxiety was positively associated with body temperature in healthy participants, indicating that the increase in activation and metabolism associated with greater anxiety increases body temperature [43]. However, this association was absent in FMS patients, suggesting lower physiological activation in response to anxiety or a dissociation between subjective and objective indicators of state anxiety.

Martinez Lavin and colleagues [19] framed FMS as a sympathetically maintained neuropathic pain syndrome. This theory is based on the hypothesis of increased tonic sympathetic activity. In its updated version, this hypothesis proposes that, in susceptible individuals, SNS hyperactivity may lead to neuropathic pain through dorsal root ganglia (which contain cell bodies of small sensory fibers and sympathetic innervations) hyperexcitability [44]. The results of our study do not support this hypothesis, given the lower tonic SC levels observed, which suggest reduced tonic sympathetic influences in FMS patients. Previous studies also found reduced sympathetic-mediated myocardial contractility at rest in FMS [17, 18]. Furthermore, our results do not support the view that fatigue and widespread pain in FM are the result of tissue ischemia produced by excessive sympathetically mediated vasoconstriction [30, 31].

Our findings of lower tonic SC and reduced SC responses may also be interpreted in the context of recent findings reporting small nerve fiber neuropathy in subsets of FMS patients. Sweating is controlled by sympathetic non-myelinated C-fibers and deficiencies therein may occur due to alterations in these nerve fibers [26, 34]. Small (intraepidermal unmyelinated nerve fibers) and even large peripheral nerve fiber neuropathy, usually in distal body parts, has been observed in a proportion of FMS patients, leading to a lower density of these fibers in comparison with healthy individuals [5, 6, 45–48]. A recent study in a large cohort of patients with FMS applied different tests for multidimensional characterization of neuropathy, observing indications of widespread small nerve fiber dysfunction and damage and identifying patient subgroups in relation to the extent and pattern of this damage [49]. The reduced small fiber density is congruent with the increased rate of neurologic symptoms like numbness, tingling, and increased heat and cold sensorial thresholds in FMS patients [50]. This peripheral neuropathy appears to exclusively affect small somatic myelinated A-delta and autonomic unmyelinated C (to smooth muscles, eccrine glands, skin hear) fibers, which can result in

sensory (pain, tingling, burning, numbness) and autonomic symptoms (dizziness, dry eyes and mouth, bladder discomfort, constipation, sexual dysfunction, decreased sweating, skin discoloration). As a result of sudomotor and vasomotor abnormalities secondary to this peripheral neuropathy, the skin over the affected areas could appear discolored, atrophic, dry, shiny, etc. Neuropathy in small somatic nerve fibers could promote pain, especially of the neuropathic type. Normal small fibers exert a filtering function by conducting only a small fraction of all incoming discharges evoked by painful inputs, while dystrophic small fibers, having lost this barrier function, unselectively conduct most of the elicited action potentials [51, 52]. Indirect methods for evaluation of small fiber neuropathy include assessment of sweating via quantitative sudomotor axon reflex testing and distal electrochemical skin conductance [26], with reduced sweating being indicative of small fiber neuropathy.

The observed decrease in nerve fiber density and axonal diameter observed in FMS, and the abnormalities in Schwann cells [46], have been reported in most chronic pain conditions, and in individuals with insulin resistance and metabolic syndrome (the main etiology of peripheral neuropathy); these may be nonspecific findings affecting only a proportion of FMS patients. The lower small fiber density may also result from the neuroplasticity and comorbidities, such as deconditioning and sedentary a lifestyle, often observed in chronic pain disorders [53].

Taken together, the results of this and previous studies suggest an overall reduction in ANS regulation at rest, both at the sympathetic (e.g., myocardial contractility, SC) and parasympathetic (HRV and BRS) level, and reduced reactivity to stress manipulations (with less marked increases seen in heart rate, blood pressure, myocardial contractility, SC, etc.) in FMS [12, 17–19]. The blunted sympathetic reactivity may imply reduced physiological resources for coping with environmental and daily life demands. The reduction in physiological mobilization would reduce the organism's resources for adjusting to situational requirements, which, according with transactional theory of stress [54], could result in inadequate coping and greater stress, thus promoting higher pain levels. The observed inverse associations between SC, both levels and responses, and pain and fatigue variables are in accordance with this notion.

The main limitation of the study is medication use, as most of patients take different combinations of medications. Due to the relatively small sample size of the study and the complex sets of medications, the analysis of the potential effect of medication on skin conductance has not been possible. The power of statistical testing would not have been sufficient to compare subgroups of patients taking and not taking the different subsets of medications. As such, the effects of medication in the observed results cannot be ruled out. Future studies with larger sample sizes and ideally with patients free of medication should analyze this issue. Some others limitations should be taken into account. First, the sample was comprised only of females, limiting the generalizability of the results. Also, differences in sample size among the measured variables resulted in differences in statistical power, which could complicate comparisons of the observed effects sizes. At the physiological level, it is necessary to take into account the high specificity of the ANS. In its role of regulating the organism's functions and adaptation to ongoing activity, the SNS input to each target organ can vary substantially, even within the same tissue (e.g., sweat glands and vascular smooth muscle in the skin) or organ (e.g., chronotropic vs. inotropic influences in the heart). Therefore, caution is necessary and the interpretation of SC results should be restricted to sweating function. Furthermore, SC cannot be taken as a reliable index of sweat function or small nerve fibers, unlike quantitative sudomotor axon reflex testing, which is a validated (invasive) method that measures sweating in response to acetylcholine administration; SC can only be considered as an indirect surrogate of sweat function. Regarding physiological measurement, asymmetry in SC and temperature has been reported in FMS, with higher SC and temperature values at the right hand [55]. SC may also

vary as a function of the type and location of electrodes (fingers vs. thenar localizations). In our study, we used disposable 1 m$^2$ electrodes localized in thenar and hypothenar parts of the right hand. Finally, it is necessary to take into account the considerable heterogeneity in psychophysiological profiles observed among FMS patients. Psychophysiological heterogeneity of FMS may partly explain the conflicting results that have been reported, suggesting the need to differentiate homogeneous subgroups of patients [56].

In conclusion, this study showed reduced SC levels, a less marked decrease in SC during the recording period, and decreased SC reactivity to a physiological manipulation, all suggesting alterations in sweating and reduced sympathetic activity to the skin. Furthermore, the absence of associations of SC with state anxiety and body temperature suggest a loss of functionality of the ANS in FMS. Together with previous findings of ANS alterations in FMS, diminished autonomic regulation in this disease would reduce the organism's ability to cope with environmental demands, thus favoring increases in stress and pain levels. Finally, the observed reduction in sweating accords with evidence of small nerve fiber neuropathy in FMS.

## Acknowledgments

This research would not have been possible without the collaboration of Fibromyalgia Association of Jaén (AFIXA).

## Author Contributions

**Conceptualization:** Gustavo A. Reyes del Paso.

**Data curation:** Pablo de la Coba.

**Formal analysis:** Pablo de la Coba.

**Funding acquisition:** Gustavo A. Reyes del Paso.

**Investigation:** Gustavo A. Reyes del Paso, Pablo de la Coba.

**Methodology:** Gustavo A. Reyes del Paso, Pablo de la Coba.

**Project administration:** Gustavo A. Reyes del Paso.

**Resources:** Gustavo A. Reyes del Paso.

**Supervision:** Gustavo A. Reyes del Paso.

**Validation:** Gustavo A. Reyes del Paso.

**Writing – original draft:** Gustavo A. Reyes del Paso, Pablo de la Coba.

**Writing – review & editing:** Gustavo A. Reyes del Paso, Pablo de la Coba.

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
