## [Decision Letter · Decision Letter 0]

17 Aug 2020

PONE-D-20-18091

Reduced activity, reactivity and functionality of the sympathetic nervous system in fibromyalgia: An electrodermal study

PLOS ONE

Dear Dr. de la Coba,

Thank you for submitting your manuscript to PLOS ONE. After careful consideration, we feel that it has merit but does not fully meet PLOS ONE’s publication criteria as it currently stands. Therefore, we invite you to submit a revised version of the manuscript that addresses the points raised during the review process.

The observations are interesting, but the queries raised need to be addressed.

We look forward to receiving your revised manuscript.

Kind regards,

Rayaz A Malik, MBChB, PhD

Academic Editor

PLOS ONE

Additional Editor Comments:

You have made a simple and important additional finding in fibromyalgia which should be presented in a clear succinct manner.

Consider the recent detailed paper showing widespread small fibre abnormalities in FMS by Evdokimov et al 2019;86: 504-516.

Is there any relationship to the severity of FMS?

Are the abnormalities noted comparable in the foot?

Journal Requirements:

2. Please update the Methods section of your manuscript to include the following:

A) Detailed inclusion and exclusion criteria for both cases and controls

B) A rationale for the diagnosis of FM with the 1990 American College of Rheumatology criteria - please state why more recent diagnostic criteria were not used.

C) Details of how participants were recruited to your study.

3. Please provide additional details regarding participant consent. In the Methods section, please ensure that you have specified what type of consent you obtained (for instance, written or verbal) and whether the ethics committee approved this consent procedure. If verbal consent was obtained please state why it was not possible to obtain written consent and how verbal consent was recorded. If your study included minors, state whether you obtained consent from parents or guardians.

Reviewers' comments:

Reviewer's Responses to Questions

**Comments to the Author**

1. Is the manuscript technically sound, and do the data support the conclusions?

Reviewer #1: Partly

2. Has the statistical analysis been performed appropriately and rigorously? 

Reviewer #1: Yes

3. Have the authors made all data underlying the findings in their manuscript fully available?

Reviewer #1: Yes

4. Is the manuscript presented in an intelligible fashion and written in standard English?

Reviewer #1: Yes

5. Review Comments to the Author

Reviewer #1: The authors describe an interesting study in patients with fibromyalgia. The data is fairly strong, although I have a couple of comments and questions for the authors.

The testing is conducted on the right hand in all patients? In some places it is listed as the right hand, in other places as the dominant hand, can this be clarified? Could the authors explain why this was the location of testing? Sweating in the hands and feet has very strong associations with emotion, so carries a significant psychological impact on testing locations. QSART testing by multiple sites on not-emotional sweating locations would have derived a cleaner result.

Skin conductance is a surrogate measure of sweat function, not a direct measure. Skin conductance is present in patients with a congenital absence of sweat glands, so there are significant limitations to what information can be extrapolated about the overall function of the autonomic system based on this limitation.

The authors don’t really describe the individuals in this study. Were any of the fibromyalgia patients (or controls) taking medications? Did they have other medical conditions? These will create a huge impact on results and need to be included.

The introduction is very long, and could be more focused. Some of the introduction could be moved to the methods or discussion section.

As there is a limited amount of data in this paper, the authors do need to try and expand the value of this manuscript by adding additional figures or creating a multi-part figure with all the relevant data from the skin conductance during the different testing periods.

6. PLOS authors have the option to publish the peer review history of their article (what does this mean?). If published, this will include your full peer review and any attached files.

Reviewer #1: No

---

## [Author Response · Author response to Decision Letter 0]

8 Sep 2020

Dear Dr. Rayaz A Malik, 

Please find attached the revision of our manuscript entitled “Reduced activity, reactivity and functionality of the sympathetic nervous system in fibromyalgia: An electrodermal study”. Changes are marked in yellow in the revision. Our responses to your comments and the reviewer´s comments are indicated below.

Sincerely

Pablo de la Coba

Responses to the editor

1. Suggestion about the recent detailed paper showing widespread small fibre abnormalities in FMS by Evdokimov et al 2019;86: 504-516.

Response: The suggested reference has been incorporated and discussed. Thanks for the suggestion.

2. Is there any relationship to the severity of FMS?

Response: We have computed correlations between skin conductance variables and intensity of pain and fatigue in the whole sample. Significant negative associations were observed for the number of pain points, sensorial pain, current pain intensity and fatigue. Data on clinical symptoms and correlations of skin conductance with symptoms severity were included in the new version of the manuscript. 

3. Are the abnormalities noted comparable in the foot?

Response: We only measure skin conductance in the hand. We used the habitual psychophysiology protocol for recording skin conductance, which entail the localization of electrodes in the hand or the fingers. 

 1. Journal Requirements: Please ensure that your manuscript meets PLOS ONE's style requirements, including those for file naming.

Response: Done. 

2. Please update the Methods section of your manuscript to include the following:

A) Detailed inclusion and exclusion criteria for both cases and controls

Response: We have completed and specified the information about inclusion and exclusion criteria for the two study groups in the Participants subsection of the method. 

B) A rationale for the diagnosis of FM with the 1990 American College of Rheumatology criteria - please state why more recent diagnostic criteria were not used.

Response: There were two main reasons to use the 1990 ACR criteria for FMS diagnosis in our study: 1) to unify diagnostic criteria, since there were several patients with a FMS diagnosis carried out before 2010; and 2) still currently in Spain, not all rheumatologists use 2010 ACR criteria, and several of them still uses the 1990 ACR criteria. We justify the use of the 1990 ACR criteria in the Participants section of the Method.

C) Details of how participants were recruited to your study.

Response: Additional information on the recruitment process has included in the Participants subsection of the method. 

3. Please provide additional details regarding participant consent. In the Methods section, please ensure that you have specified what type of consent you obtained (for instance, written or verbal) and whether the ethics committee approved this consent procedure. If verbal consent was obtained please state why it was not possible to obtain written consent and how verbal consent was recorded.

Response: We used a written consent. Additional details regarding participants consent and ethic committee approval have been added to the ending of Participant subsection of the method.

Response: The research data of the study are available to the public via the repository Open Science Framework (OSF: https://osf.io/3uk4e/).

Responses to the reviewer

The testing is conducted on the right hand in all patients? In some places it is listed as the right hand, in other places as the dominant hand, can this be clarified? Could the authors explain why this was the location of testing? Sweating in the hands and feet has very strong associations with emotion, so carries a significant psychological impact on testing locations. QSART testing by multiple sites on not-emotional sweating locations would have derived a cleaner result.

Response: Testing was performed in all participants in the right hand. We have clarified this issue in the current version of the manuscript. We used for skin conductance recording the usual psychophysiology procedure for the measurement of levels and responses in which electrodes where located in the hands (the hands join with the feet are the body parts with more sweeting glands; Dawson, Schell, and Filion, 2016). In the protocol for the recording of electrochemical conductance, the only used body localizations are the hand and feet. We do not know studies in which skin conductance were recorded in other body parts. 

Skin conductance is a surrogate measure of sweat function, not a direct measure. Skin conductance is present in patients with a congenital absence of sweat glands, so there are significant limitations to what information can be extrapolated about the overall function of the autonomic system based on this limitation.

Response: We are completely agree with the reviewer and we further emphasize this consideration in the limitation subsection of the discussion.

The authors don’t really describe the individuals in this study. Were any of the fibromyalgia patients (or controls) taking medications? Did they have other medical conditions? These will create a huge impact on results and need to be included.

Response: Based on the suggestion of the reviewer, and to better characterize our patients sample, we have included in Table 1 some clinical data of participants, including medication use. Most of the patients take different sets of medications, including antidepressants, anxiolytic, non-opioids analgesic and opioids. 

Unfortunately, due to the relatively small sample size of the study and the complex sets of used medications, the analysis of the potential effect of medications on skin conductance has been not possible. The power of statistical testing would not have been sufficient to compare the different subgroups of patients taking and not taking the different subsets of medications. As such, the effects of medication in the observed results cannot be ruled out. We have acknowledged this as the main limitation of the study in the revised Discussion as follow:

“The main limitation of the study is medication use, as most of patients take different combinations of medications. Due to the relatively small sample size of the study and the complex sets of medications, the analysis of the potential effect of medication on skin conductance has not been possible. The power of statistical testing would not have been sufficient to compare subgroups of patients taking and not taking the different subsets of medications. As such, the effects of medication in the observed results cannot be ruled out. Future studies with larger sample sizes and ideally with patients free of medication should analyze this issue.”

The introduction is very long, and could be more focused. Some of the introduction could be moved to the methods or discussion section.

Response: The introduction section has been substantially shortened (several paragraphs have been omitted) and focused in the study aim.

As there is a limited amount of data in this paper, the authors do need to try and expand the value of this manuscript by adding additional figures or creating a multi-part figure with all the relevant data from the skin conductance during the different testing periods.

Response: Following the reviewer suggestion, we have included a new figure with the data of the response to the breathing manipulation and a table displaying the correlations. Furthermore, a table showing the correlations between skin conductance variables and pain and fatigue variables has been included. 

Dawson ME, Schell AM, Filion DL. The Electrodermal System. In: Cacioppo JT, Tassinary LG, Berntson GG. Handbook of Psychophysiology. Cambridge: University Press, Cambridge; 2016. pp. 217-243.

---

## [Editor Report · Decision Letter 1]

9 Oct 2020

Reduced activity, reactivity and functionality of the sympathetic nervous system in fibromyalgia: An electrodermal study

PONE-D-20-18091R1

Dear Dr. de la Coba,

We’re pleased to inform you that your manuscript has been judged scientifically suitable for publication and will be formally accepted for publication once it meets all outstanding technical requirements.

Kind regards,

Rayaz A Malik, MBChB, PhD

Academic Editor

PLOS ONE

Additional Editor Comments (optional):

All main comments have been addressed
---

## [Editor Report · Acceptance letter]

16 Oct 2020

PONE-D-20-18091R1 

Reduced activity, reactivity and functionality of the sympathetic nervous system in fibromyalgia: An electrodermal study 

Dear Dr. de la Coba:

I'm pleased to inform you that your manuscript has been deemed suitable for publication in PLOS ONE. Congratulations! Your manuscript is now with our production department. 

Kind regards, 

on behalf of

Professor Rayaz A Malik 

Academic Editor

PLOS ONE